# Are You Getting What You Pay For? Auditing Model Substitution in LLM APIs

## Abstract

Commercial Large Language Model (LLM) APIs create a fundamental trust problem: users pay for specific models but have no guarantee that providers deliver them faithfully. Providers may covertly substitute cheaper alternatives (e.g., quantized versions, smaller models) to reduce costs while maintaining advertised pricing. We formalize this *model substitution* problem and systematically evaluate detection methods under realistic adversarial conditions. Our empirical analysis reveals that software-only methods are fundamentally unreliable: statistical tests on text outputs are query-intensive and fail against subtle substitutions, while methods using log probabilities are defeated by inherent inference nondeterminism in production environments. We argue that this verification gap can be more effectively closed with hardware-level security. We propose and evaluate the use of Trusted Execution Environments (TEEs) as one practical and robust solution. Our findings demonstrate that TEEs can provide provable cryptographic guarantees of model integrity with only a modest performance overhead, offering a clear and actionable path to ensure users get what they pay for.

## 1 Introduction

Large Language Models (LLMs) have demonstrated remarkable capabilities, leading to widespread adoption through cloud-based APIs (OpenAI, 2022; Anthropic, 2023; Google, 2023; TogetherAI, 2023). Enterprises and researchers select and pay for specific models (e.g., Llama 405B vs 70B) based on advertised capabilities, performance on leaderboards, and expected behavior (Touvron et al., 2023; Bai et al., 2023; Mistral, 2023). However, this black-box access model operates on an implicit assumption of trust: that the provider will faithfully serve the requested model. The immense computational cost of hosting state-of-the-art LLMs creates a powerful economic incentive for providers to violate this trust through *model substitution*—covertly replacing the advertised model with a cheaper, less powerful alternative.

This substitution threat is not hypothetical. A provider might swap a flagship model for a smaller variant or a heavily quantized version (e.g., INT8/FP8) to reduce GPU costs and increase profit margins (Gao et al., 2024; Sun et al., 2024b). Substitutions can also occur for operational reasons, such as rerouting traffic from an overloaded high-end model to a less-utilized, lower-tier one (e.g., a version fine-tuned on different data). While the intent may not always be malicious, any undisclosed substitution breaks the service agreement, compromises the reliability of dependent applications, hinders reproducible research, and invalidates the benchmark results. Furthermore, subtle changes from quantization or fine-tuning might inadvertently affect model safety or introduce biases (Hong et al., 2024). Imagine a researcher relying on a specific model's advertised capabilities for scientific analysis, only to unknowingly receive results from a substituted, less capable model, potentially invalidating their findings.

For a user or auditor, verifying the model behind a black-box API is fundamentally challenging. The interaction is typically limited to sending prompts and receiving text (and optionally, token log probabilities (Carlini et al., 2024)), with no direct access to the model's weights or the underlying infrastructure. This *information asymmetry* heavily favors the provider. A sophisticated provider can exploit this by employing active countermeasures, such as detecting and routing benchmark queries to the genuine model while serving a substitute to regular traffic, or by randomly mixing outputs from different models to obscure the statistical signal of the substitution.

This paper confronts the critical problem of auditing model substitution in black-box LLM APIs. We formalize the verification task, analyze the limitations of existing software-based methods under realistic adversarial conditions, and present a robust solution. Our central thesis is that the inherent ambiguity and nondeterminism of software-level signals make them insufficient for reliable auditing. Instead, we argue that hardware-backed attestation via Trusted Execution Environments (TEEs) is the only currently viable mechanism to achieve strong, efficient, and provable integrity for LLM APIs. Our contributions are:

- We formalize the problem of LLM model substitution detection from the perspective of a black-box user/auditor.
- We design and analyze practical adversarial scenarios for model substitution, including quantization, randomized substitution, and benchmark evasion.
- We empirically evaluate the effectiveness of existing detection techniques, showing that text-based statistical tests lack power against subtle or randomized substitutions, while metadata-based methods like log probability comparison are defeated by production-level inference nondeterminism.
- We propose and evaluate TEEs as a practical and deployable solution, demonstrating that they uniquely provide provable model integrity in black-box APIs. We show that TEEs offer strong security guarantees with only a modest performance cost, establishing them as the most actionable path toward substitution-resistant LLM services.

## 2 RELATED WORK

**LLM API monitoring and auditing.**  Several studies have monitored the behavior and performance of commercial LLM APIs over time. Chen et al. (2023) track changes in ChatGPT's capabilities, highlighting behavioral drift, while Eyuboglu et al. (2024) characterize updates to API-accessed ML models. Closer to our work, Gao et al. (2024) test API output text distributions using MMD; concurrently, Zhu et al. (2025) introduce a rank-based uniformity test as an alternative distributional check. Other approaches include direct auditing via identity-style prompting (Huang et al., 2025) and predicting black-box LLM performance using self-queries (Sam et al., 2025). When model internals are available, TopLoc (Ong et al., 2025) uses locality-sensitive hashing over intermediate activations to produce proofs of correct execution. While these methods provide a foundation for auditing, their robustness against determined adversaries and in realistic, non-deterministic production environments (He & Lab, 2025) remains an open question, which we address in this work.

**Detecting LLM-generated text.**  Detection of LLM-generated text has been extensively studied, including post-hoc methods and proactive watermarking techniques (Yang et al., 2023; Ghosal et al., 2023). Zero-shot detection approaches use stylistic features or model-specific "fingerprints" to distinguish AI-generated content without specialized training data (Mitchell et al., 2023; Bao et al., 2023; Yang et al., 2024). Trained classifiers (Hu et al., 2023; Sun et al., 2025) utilize datasets from various sources to differentiate model outputs. Although effective against stylistically distinct models, these approaches may struggle with subtle substitutions within the same model family. LLM watermarking techniques embed hidden signals in outputs to trace content ownership (Kirchenbauer et al., 2023; Zhao et al., 2024a;b), but they are provider-centric and not intended for end-user verification of model *identity*, particularly as the same watermark might span different backend models.

**Verifiable computation for ML.** Cryptographic and hardware-based techniques can provably verify ML inference. Zero-Knowledge Proofs (ZKPs) allow proof of correct inference without revealing inputs or model weights but face substantial computational costs for large LLMs, making them impractical for real-time APIs today (Sun et al., 2024a; Xie et al., 2025). In contrast, Trusted Execution Environments (TEEs) provide hardware-level guarantees of integrity and confidentiality with much lower overhead (NVIDIA, 2023). Our work builds on this insight, arguing that TEEs represent the most mature and practical path toward verifiable LLM inference in the *current* ecosystem.

## 3 PROBLEM FORMULATION AND THREAT MODELS

### 3.1 PROBLEM FORMULATION

Consider an LLM API service with three entities:

- **User/Auditor:** Use LLM services for tasks with input prompts $x$ drawn from a distribution $\pi(x)$.
- **Service provider:** Offers access to an LLM via a black-box API. The provider claims to serve a specific target model $M_{\text{spec}}$ but may substitute it with $M_{\text{alt}}$.
- **Target model ($M_{\text{spec}}$):** The advertised model, identified by its exact weight checkpoint $W_{\text{spec}}$.

**Model substitution** occurs when the provider uses model weights $W_{\text{actual}} \neq W_{\text{spec}}$ without disclosure. Distributional differences (e.g., $P_{\text{actual}}(y|x)$ vs. $P_{\text{spec}}(y|x)$) are behavioral consequences of such weight changes, not the substitution definition itself. Substitutions include: (1) smaller models (e.g., 7B vs 70B), (2) quantized variants (INT8 vs FP16) of $M_{\text{spec}}$, (3) fine-tuned or updated versions with altered training data or objectives, or (4) entirely different model families.

**Open-source vs. proprietary access.** The user's ability to audit depends on the type of model and API. For *open-source models*, an auditor can run $M_{\text{spec}}$ locally to obtain reference log probabilities or greedy outputs. For *proprietary models* (e.g., GPT-4, Claude-4), the auditor only has black-box access, and therefore must rely solely on behavioral evidence as a proxy for underlying weights.

**Auditing goal.** The user/auditor aims to determine whether the provider is faithfully using the specified model checkpoint. Let $M_{\text{spec}}$ denote the advertised model, defined by its exact weight checkpoint, and let $M_{\text{actual}}$ denote the model actually used by the provider. Because the auditor only observes samples $(x, y)$ from the API, any distributional objects (e.g., $P_{\text{spec}}$, $P_{\text{actual}}$) are indirect behavioral proxies inferred from finite samples rather than direct evidence of model identity. The audit objective is to test:

$$H_0 : M_{\text{actual}} = M_{\text{spec}} \quad \text{(Honest provider)}$$

against

$$H_1 : M_{\text{actual}} \neq M_{\text{spec}} \quad \text{(Substitution occurred)}.$$

Later sections evaluate distribution-based methods (benchmarks, MMD) as proxy tests for this model weight-level hypothesis. These proxies are inherently sensitive to inference-time nondeterminism and decoding settings and therefore cannot certify model equality, but they can still provide useful behavioral signals: substantial distributional drift is a necessary yet not sufficient condition for detecting many forms of model substitution.

## 3.2 Adversarial attack scenarios

**Quantization substitution.** The provider replaces the full-precision target model $M_{\text{spec}}$ with a quantized version (e.g., INT8, FP8, NF4). Quantization significantly reduces memory footprint and often accelerates inference, lowering costs. While preserving much of the model's capabilities, it slightly alters the output distribution $P_{\text{alt}}(y|x)$. The attack relies on this distributional difference being too small for naive detection methods to pick up, especially with limited samples.

**Randomized model substitution.** To make detection harder, the provider can route a query to a cheaper substitute $M_{\text{alt}}$ with probability $p$ and to the specified model $M_{\text{spec}}$ with probability $1 - p$. The observed distribution is $P_{\text{mixed}}(y \mid x) = p \cdot P_{\text{alt}}(y \mid x) + (1 - p) \cdot P_{\text{spec}}(y \mid x)$. As $p \to 0$ (low substitution rate), $P_{\text{mixed}}$ approaches $P_{\text{spec}}$, making sampling-based detection increasingly difficult. A sophisticated provider might decrease $p$ when traffic appears audit-like and increase it on ordinary traffic.

**Benchmark evasion (cached/routed output).** This attack targets verification methods relying on known, fixed prompts, such as benchmark datasets or identity queries. The provider detects likely audit queries (e.g., via hashing or embedding similarity) and routes them to a genuine $M_{\text{spec}}$ instance or returns cached outputs and metadata. Ordinary traffic is served by $M_{\text{alt}}$.

**Limiting information disclosure.** Providers might proactively limit the information exposed via APIs (e.g., removing full logits, restricting top-$k$ log probabilities) after demonstrations that such features can leak sensitive details (Carlini et al., 2024; Finlayson et al., 2024).

## 4 Model verification techniques and robustness analysis

LLM service providers offer a wide range of interfaces, from basic text-only web chats to advanced APIs that allow control over decoding parameters and access to token log probabilities (Table 1 and

| Service Provider | Open Source Models | Decoding Parameters | Logprobs Output |
|---|---|---|---|
| Anyscale | Yes | Full Control | Yes |
| Together.ai | Yes | Full Control | Yes |
| Hugging Face | Yes | Full Control | Top 5 |
| AWS Bedrock | Yes | Full Control | Yes |
| Nebius AI | Yes | Full Control | Yes |
| Vertex AI | Yes | Full Control | Top 5 |
| Mistral | Yes | Partial Control | No |
| DeepSeek | Yes | Full Control | Top 20 |
| OpenAI | No | Partial Control | Depends on model |
| Cohere | No | Full Control | Top 1 |
| Anthropic | No | Full Control | No |

Table 1: Transparency and control across LLM API providers (September 2025). "Full Control" implies typical parameters like temperature, top-p, top-k, etc. Log probability availability and limits vary by model/version.

Table 5). The level of information disclosure significantly constrains which verification techniques can be applied. In this section, we systematically evaluate various auditing methods across different access levels: (1) text-only output, (2) output plus metadata (e.g., log probabilities, activations), and (3) provider-supplied integrity (e.g., TEEs in Section 4.3). For each technique, we detail the methodology, describe our experimental setup for evaluating its robustness against the adversarial scenarios outlined in Section 3.2, and analyze its effectiveness.

## 4.1 TEXT-OUTPUT-BASED VERIFICATION

### 4.1.1 TEXT CLASSIFIER

**Method.** Sun et al. (2025) use a trained classifier to predict the source model of generated samples, exploiting stylistic "fingerprints" unique to each model. Given a dataset of model completion labeled with model, the classifier is trained to predict the label using cross-entropy loss.

**Attack: Quantization substitution.** We test whether classifiers could distinguish full-precision from quantized variants of Llama-3.1-70B, Gemma-2-9B, Mistral-7B-v0.3, and Qwen2-72B. Using Ultra-Chat prompts (temperature 0.6), we train on 10k samples per variant (with 1k validation) and evaluate classifiers with BERT, T5, GPT-2, and LLM2Vec embeddings.

| Model | BERT Acc | T5 Acc | GPT2 Acc | LLM2Vec Acc |
|---|---|---|---|---|
| Llama3-70B-Instruct-FP8 | 50.55 | 50.10 | 50.45 | 49.90 |
| Llama3-70B-Instruct-INT8 | 51.60 | 49.90 | 51.30 | 50.25 |
| Gemma2-9b-it-FP8 | 49.95 | 50.30 | 51.20 | 49.80 |
| Gemma2-9b-it-INT8 | 49.00 | 49.70 | 51.65 | 49.55 |
| Mistral-7b-v3-Instruct-FP8 | 50.55 | 49.75 | 48.70 | 48.75 |
| Mistral-7b-v3-Instruct-INT8 | 49.70 | 50.75 | 50.50 | 51.15 |
| Qwen2-72B-Instruct-FP8 | 50.05 | 50.25 | 48.75 | 49.80 |
| Qwen2-72B-Instruct-INT8 | 50.75 | 50.55 | 49.45 | 50.20 |

Table 2: Binary classification accuracy between outputs of quantized and original models on UltraChat (Ding et al., 2023). Accuracies near 50% indicate failure to differentiate.

**Analysis.** Classification accuracies stay near chance across all model pairs and embedding backbones (Table 2), leaving us unable to reject $H_0$ that $M_{spec}$ and $M_{alt}$ produce indistinguishable outputs; deviations from 50% can be attributed to sampling noise. The result is consistent across families, parameter sizes, and INT8/FP8 variants, and persists under alternative train/validation splits. In contrast to Sun et al. (2025), who report around 98% cross-model and 80% cross-model size accuracy, the signal here is too weak to separate quantized from full-precision variants. We therefore find that text-based classifiers, even with strong encoders (BERT/T5/GPT-2/LLM2Vec), are ineffective for detecting quantization-based substitution under realistic sampling budgets.

### 4.1.2 IDENTITY PROMPTING

**Method.** Prompt the model to disclose its identity (e.g., "Who are you?", "Who developed you?") and string-match against known identifiers (family, developer), following the simple classifier setup in Huang et al. (2025).

**Attack: Quantization substitution.** We test whether quantization changes self-identification: for Llama3.1-70B-Instruct, Gemma2-9b-it, Mistral-7b-v3-Instruct, and Qwen2-72B-Instruct, we issue

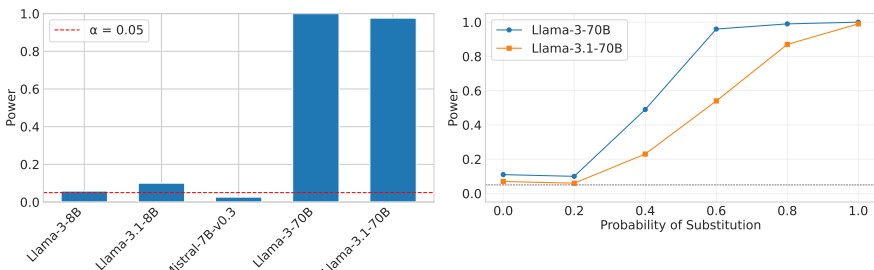

Figure 1: Statistical power of MMD tests. (Left) Full-substitution with quantized models: power remains near $\alpha$ for 7B/8B models, while tests on 70B models yield power close to one. (Right) Mixed-substitution with quantized models: Power as a function of substitution probability $p$ for larger models, showing how mixtures of quantized and full-precision outputs affect detectability.

500 identity queries per model/variant at temperature 0.6 and matched family (e.g., "Llama") and developer (e.g., "Meta").

**Attack: System prompt override.** In this setting, we explicitly set the system prompt to mislabel the model (e.g., "*You are LLaMA, an AI assistant developed by Meta AI*") and then re-ran identity queries on Gemma2-9b-it, Mistral-7b-v3-Instruct, and Qwen2-72B-Instruct.

**Analysis.** The results for quantization substitution remain consistent: models typically disclose only broad identifiers such as the family or developer, and rarely reveal specific details like parameter count or quantization status. Queries asking for direct specifications are successful in less than 5% of cases, often resulting in refusals or incorrect responses from the model. When the system prompt is overridden, most models fail to correctly identify themselves, with accuracy dropping to nearly 0%. Instead, they often adopt the identity given by the system prompt, although some contradictions occasionally occur (e.g., restating the true developer). Overall, these findings indicate that identity prompting is ineffective for detecting substitutions within the same family and is highly sensitive to changes in the prompt, making it an unreliable method for detecting substitutions in practice.

### 4.1.3 Model equality testing

**Method.** When only text outputs are available, one can test whether two models induce the same distribution over completions on a fixed set of prompts by applying a Maximum Mean Discrepancy (MMD) test (Gao et al., 2024). Let $P(z \mid x)$ and $Q(z \mid x)$ denote the distributions over completions produced by the two candidate models for a given prompt $x$, and let $z$ and $z'$ denote individual completions sampled from these conditional distributions.

$$\mathrm{MMD}(P, Q) = \mathbb{E}_{z,z' \sim P}[k(z, z')] + \mathbb{E}_{z,z' \sim Q}[k(z, z')] - 2\mathbb{E}_{z \sim P, z' \sim Q}[k(z, z')],$$

with the Hamming kernel $k_{\mathrm{hamming}}(z, z') = \sum_{i=1}^{L} \mathbf{1}\{z_i = z_i'\}$. We estimate significance by permutation testing: repeatedly shuffle completions between groups, recompute MMD, and reject $H_0$ if the observed statistic exceeds the $\alpha$-quantile of this null distribution.

**Attack: Quantization substitution.** We first compare outputs from $M_{\mathrm{spec}}$ (full precision) and $M_{\mathrm{alt}}$ (INT8 quantized) models. For the small-model setting in Figure 1 (left), we use Meta-Llama-3-8B-Instruct and Mistral-7B-Instruct, generate completions up to $L = 50$ tokens with decoding temperature $T = 1.0$, and sample $N = 10$ completions per prompt for 25 multilingual Wikipedia prompts. We conduct 100 Monte Carlo simulations with 1000 permutations per test. Power is computed as the probability of rejecting $H_0$:$P_{\mathrm{actual}}{=}P_{\mathrm{spec}}$ when the entire output distribution came from $M_{\mathrm{alt}}$.

**Attack: Randomized substitution.** To simulate partial replacement, we construct mixtures of completions from $M_{\mathrm{spec}}$ and $M_{\mathrm{alt}}$ with substitution probability $p \in \{0.0, 0.2, \ldots, 1.0\}$. We then repeat the same MMD test procedure to estimate how power changes as $p$ increases. This setup models a provider routing only a fraction of queries to the substituted model while serving the remainder from the specified one.

In figure 1, in the full-substitution setting shown in the left panel, the statistical power stays close to the significance level $\alpha$, indicating that quantization on small models introduces only a very small shift in the output distribution and is extremely hard to detect. In contrast, tests on large models yield power close to one, so we focus on the more subtle mixed-substitution setting. The right panel shows that power increases smoothly with the substitution probability $p$ and reaches high values when $p = 1$, illustrating that the test becomes effective once the substituted model contributes substantially to the output distribution. Moreover, inference nondeterminism across providers can inflate MMD values even for identical models, risking false positives. Gao et al. (2024) show that, when enough samples are available, an MMD test with a simple string kernel is statistically powerful and can flag commercial endpoints whose output distributions differ from reference weights. However, both their analysis and ours highlight an important limitation: the test only certifies that the API's distribution $Q$ differs from the reference $P$; it cannot distinguish deliberate model substitution from other sources of distributional shift (e.g., inference nondeterminism or differences in decoding setting).

### 4.1.4 BENCHMARK-BASED DETECTION

**Method.** An auditor may attempt benchmark-based detection (e.g., Chen et al. (2023)) by querying standardized benchmarks where baselines are well established. We evaluate four tasks: MMLU (Hendrycks et al., 2020), GSM8K (Cobbe et al., 2021), MATH (Hendrycks et al., 2021), and GPQA Diamond (Rein et al., 2024). At decoding temperature $T=0.5$, we repeat evaluation 10 times and report mean accuracy and standard deviation (Table 3). MMLU is scored by restricting completions to multiple-choice tokens and averaging over 100 runs per query; free-form numeric tasks (GSM8K, MATH) are scored by regex matching or the `math-verify` library; GPQA is scored by multiple-choice accuracy.

**Attack 1: Quantization substitution.** We compare quantized and full-precision variants across all four benchmarks. As shown in Table 3, differences are small and typically within one standard deviation across repeated runs. This suggests that quantized models perform nearly indistinguishably from their full-precision counterparts, limiting the power of benchmark-only detection.

**Attack 2: Hiding temperature parameter.** We also evaluate benchmark accuracy under varying decoding temperature (Figure 2). For $T \leq 1$, accuracy variance across runs is low, and single-run results are stable. However, if the provider does not disclose or allow control of temperature, observed accuracy drops could be misattributed—either to hidden parameter choices or to substitution—reducing audit reliability.

| Model | MMLU | GSM8K | MATH | GPQA Diamond |
|---|---|---|---|---|
| Meta-Llama-3-8B-Instruct | $62.69 \pm 0.18$ | $61.14 \pm 3.47$ | $20.65 \pm 3.43$ | $22.62 \pm 0.22$ |
| Meta-Llama-3-8B-Instruct-FP8 | $62.43 \pm 0.26$ | $60.90 \pm 4.10$ | $14.91 \pm 2.49$ | $20.14 \pm 0.24$ |
| Meta-Llama-3-70B-Instruct | $78.05 \pm 0.08$ | $88.06 \pm 1.44$ | $35.69 \pm 1.33$ | $29.60 \pm 0.51$ |
| Meta-Llama-3-70B-Instruct-FP8 | $77.88 \pm 0.13$ | $87.35 \pm 1.24$ | $35.75 \pm 1.16$ | $33.12 \pm 0.30$ |
| Gemma-2-9b-it | $71.86 \pm 0.08$ | $81.80 \pm 1.35$ | $33.41 \pm 0.28$ | $28.64 \pm 2.97$ |
| Gemma-2-9b-it-FP8 | $71.92 \pm 0.11$ | $79.41 \pm 1.14$ | $32.53 \pm 0.34$ | $27.81 \pm 3.22$ |
| Qwen2-72B-Instruct | $82.18 \pm 0.08$ | $86.72 \pm 1.00$ | $37.39 \pm 1.41$ | $29.93 \pm 2.71$ |
| Qwen2-72B-Instruct-FP8 | $81.98 \pm 0.08$ | $86.82 \pm 0.97$ | $37.67 \pm 1.39$ | $31.08 \pm 1.96$ |
| Mistral-7B-Instruct-v0.3 | $59.15 \pm 0.10$ | $35.90 \pm 4.54$ | $8.94 \pm 1.24$ | $21.60 \pm 0.17$ |
| Mistral-7B-Instruct-v0.3-FP8 | $58.77 \pm 0.13$ | $32.20 \pm 4.02$ | $7.68 \pm 1.23$ | $22.72 \pm 0.19$ |

Table 3: Mean $\pm$ standard deviation of benchmark results on different models at temperature $T = 0.5$.

**Analysis.** Figure 2 shows that benchmark accuracy degrades with higher $T$, and the size of this drop is comparable to quantization-induced differences. This means auditors testing $H_0 : P_{\text{actual}} = P_{\text{spec}}$ cannot distinguish whether deviations from baseline arise from $M_{\text{alt}}$ substitution or from hidden parameter settings. Caching further complicates detection: if providers cache outputs for known benchmark queries, apparent accuracy may remain aligned with $M_{\text{spec}}$ even under substitution, suppressing deviations. Overall, benchmark-based detection can expose large-scale substitution, but smaller changes (e.g., quantization) are hidden by natural variability, and confounds like temperature or caching make $H_0$ rejection unreliable in practice.

### 4.1.5 GREEDY DECODING OUTPUTS

**Method.** We test whether greedy decoding, which removes sampling randomness, yields reproducible completions across settings. Beyond substitution attacks, we also investigate how greedy decoding behaves in real-world auditing conditions, where inference is accessed only through APIs. This lets us assess not just sensitivity to quantization or fine-tuning, but also whether greedy decoding (and text-based verification more broadly) is robust when applied in deployment settings.

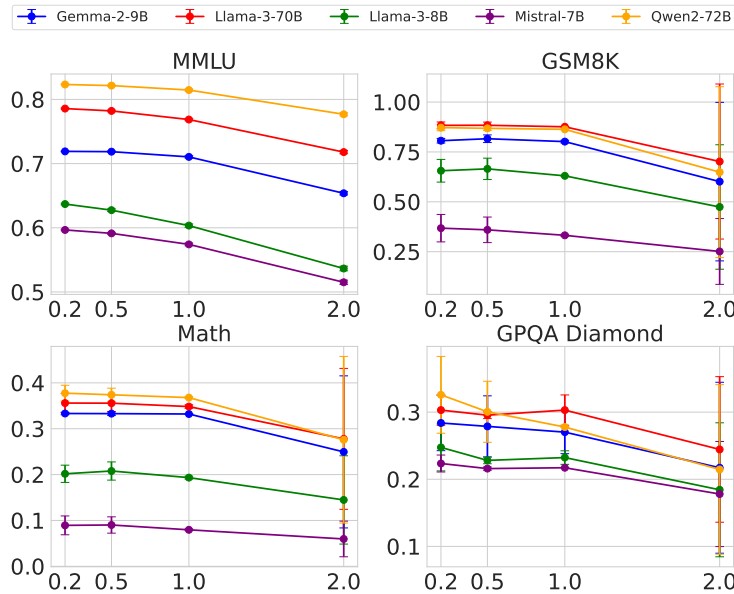

Figure 2: Benchmark accuracy versus decoding temperature across models and tasks.

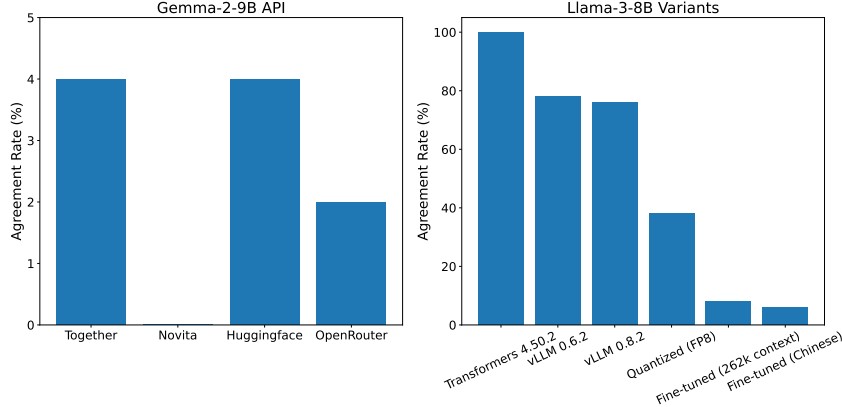

Figure 3: Greedy decoding agreement rate with local baseline (Transformers 4.40.0, H100). Left: Gemma-2-9B APIs across providers. Right: Llama-3-8B variants including quantized and fine-tuned models.

Using 50 UltraChat queries, we compare local inference runs (Transformers 4.40.0 on H100 GPUs) against API providers' outputs on the same Gemma-2-9B model. We also assess self-consistency across repeated API calls. Agreement is measured by exact match of the first $k$ tokens, with $k = 20$ for API–local comparisons (larger $k$ push agreement rates close to 1) and $k = 100$ for local comparisons among full-precision, quantized, and fine-tuned variants.

**Attack: Quantization and fine-tuned substitution.** For Llama-3-8B, we compare greedy outputs from the reference model against its quantized (FP8) and fine-tuned variants (e.g., models trained for other context length or on domain-specific corpora). While local inference across frameworks is largely consistent, both quantization and fine-tuning introduce significant token-level discrepancies, reducing agreement with the baseline sequence. This shows that greedy decoding can, in theory, detect within-family substitutions when the environment is controlled.

**Analysis.** Figure 3 shows that greedy decoding does not ensure reproducibility: even with sampling disabled, agreement across provider APIs for Gemma-2-9B is below 5%, and agreement for fine-tuned or quantized Llama-3-8B variants drops well below the baseline. Under $H_0 : P_{\text{actual}} = P_{\text{spec}}$, greedy outputs should match exactly, but observed divergence is often larger than that induced by quantization or fine-tuning. This reflects inference nondeterminism (He & Lab, 2025), whose sources (e.g., backend variability in kernels, tokenization, or caching) are discussed in the next sec-

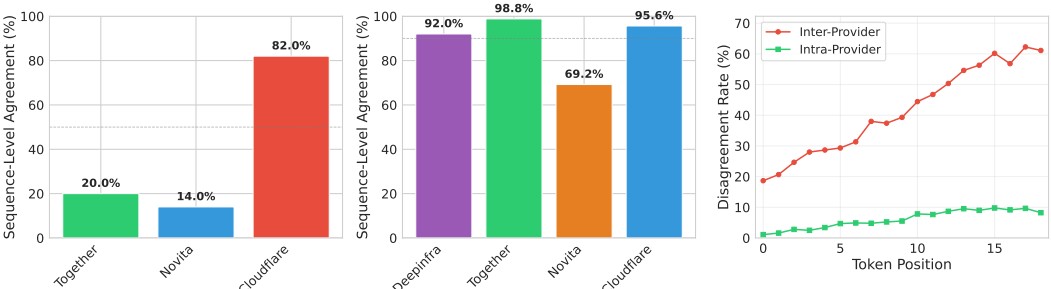

Figure 4: Greedy decoding agreement for Llama-3-8B across API providers. (Left) Average inter-provider sequence agreement rate across 3 providers vs. DeepInfra baseline. (Middle) Intra-provider agreement rate over 5 repeated queries per prompt. (Right) Average disagreement rates by token position.

tion. The key implication is that while greedy decoding can detect substitution in a controlled local environment, it fails to do so in realistic auditing settings where backend nondeterminism dominates, making $M_{\text{spec}}$ and $M_{\text{alt}}$ indistinguishable.

**API provider empirical study.** We sample 50 prompts from UltraChat (Ding et al., 2023) and generate greedy completions from several reputable API providers. For inter-provider agreement, each provider (Together, Novita, Cloudflare) is queried once per prompt and compared against a DeepInfra baseline. For intra-provider determinism, each API is queried 5 times per prompt. We restrict comparisons to $k=20$ tokens due to credit limits and prior observations that most disagreements emerge early. Sequence-level agreement captures exact-match consistency, while token-level disagreement rates quantify how differences accumulate across the generated sequence.

**Analysis.** Intra-provider agreement is relatively stable, but inter-provider agreement is notably low. Despite matching all user-facing decoding parameters, Novita achieves only 14% sequence-level agreement with DeepInfra, and disagreement rises sharply within the first 20 tokens. This indicates that inference-time nondeterminism and undisclosed inference-time setup can produce substantial drift even when providers appear to run the same model with identical settings. Our queries occur within a short continuous window; over longer periods with varying traffic, discrepancies are likely larger. Assuming these providers are honest, this variance implies that distribution-based audits (including MMD) face significant noise and are unreliable as primary tests for detecting weight-level substitution.

### 4.1.6 LOG PROBABILITY VERIFICATION

**Method.** In theory, logprobs comparison can be used to verify whether the served model matches a claimed reference: per-token log probabilities should align closely. In practice, the reliability of this approach depends on the magnitude of nondeterminism affecting the logprobs.

**Evaluation.** To assess stability, we compare token-level log probabilities from greedy decoding on UltraChat queries, across multiple inference frameworks (vLLM (Kwon et al., 2023) vs. Hugging Face Transformers (Wolf et al., 2020)), GPU types (H100, A100), and software versions. Figure 5 shows that even for the same model, logprob traces for the first 20 tokens can diverge across stacks. On the subset of completions where the generated tokens agreed, we further compare the magnitude of the assigned log probabilities and find minor variations across environments. Additional examples are given in Appendix C.

**Analysis and weakness.** Logprob traces often appear nearly identical across token positions in controlled settings, and the magnitude differences are still distinguishable when compared to quantized or fine-tuned variants. In production, however, batching and heterogeneous backends introduce much larger variance. Recent work (He & Lab, 2025) shows that a key source of this instability is the lack of batch invariance in kernels such as RMSNorm, matrix multiplication, and attention: the

same request may yield different results depending on how many other requests are batched concurrently or how prefill/chunking is scheduled. Even at $T = 0$, greedy decoding can diverge after only tens of tokens, making logprob distributions very noisy. Prototype batch-invariant kernels exist but incur throughput costs, so current APIs typically remain nondeterministic. For auditors, this means logprob-based detection is vulnerable to false alarms unless requests are isolated or providers explicitly expose batch-invariant execution. In addition, this method requires a reference instance of $M_{spec}$ and API access to log probabilities. Stronger auditing methods that attempt to reconstruct embedding subspaces (Appendix C) are generally infeasible under current API non-disclosure of full logprobs.

## 4.2 INTERNAL ACTIVATIONS VERIFICATION

**Method.** TopLoc (Ong et al., 2025) introduces a locality-sensitive hashing scheme over intermediate activations to produce compact proofs of correct execution. Here, locality-sensitive hashing (LSH) refers to mapping high-dimensional activation tensors into small fingerprints such that similar activations yield the same hash with probability, whereas substantial changes to the activations are unlikely to collide and therefore become detectable. Instead of storing full tensors, TopLoc extracts the top-$k$ largest activation values and indices and encodes them as a polynomial congruence, yielding proofs that are robust to GPU nondeterminism and algebraic reorderings. This allows compact verification (about 100 bytes per tens of tokens) while maintaining accurate detection under adversarial substitution.

**Analysis and weakness.** TopLoc can detect most model substitutions (e.g., quantized variants, fine-tuned models) even under different software and hardware settings, aligning with the behaviors observed in logprobs testing. However, verification still requires the auditor to recompute the same hidden activations, i.e., to run $M_{spec}$

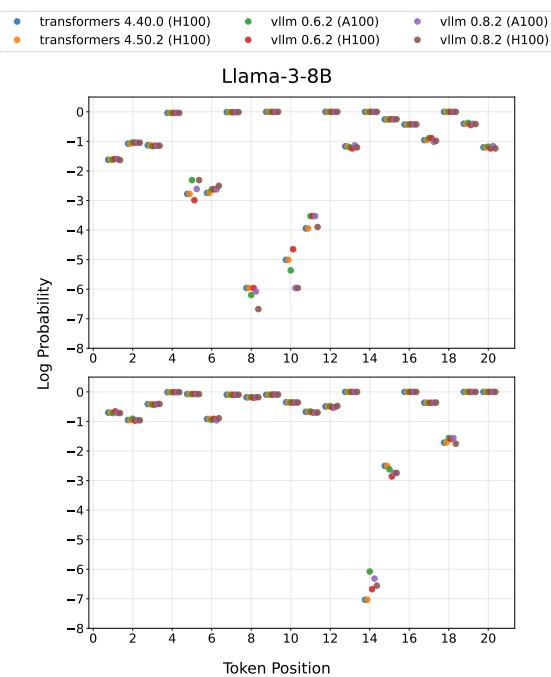

Figure 5: Log probabilities for the first 20 shared tokens under greedy decoding for UltraChat queries across frameworks/hardware.

locally or rely on an attested recomputation (e.g., a TEE), so it is infeasible without attestation for proprietary models. In addition, batching variance and speculative decoding could still remain problematic, as both can introduce enough instability in the forward pass or generated tokens to disrupt verification.

## 4.3 HARDWARE-ASSISTED VERIFICATION: TRUSTED EXECUTION ENVIRONMENT (TEE)

**Method.** Trusted Execution Environments (TEEs), such as those provided by NVIDIA's Confidential Computing on Hopper and Blackwell GPUs (NVIDIA, 2023), offer a paradigm shift. A TEE creates a hardware-isolated enclave where both the model weights and the inference code are protected from the host system. The TEE can produce a cryptographic attestation report that includes measurements (hashes) of the loaded model and execution code. An end-user can cryptographically verify this report to gain absolute certainty that the intended model is running unmodified within the specified, trusted software environment.

**Evaluation.** The primary concern with TEEs is performance overhead. We benchmarked a vLLM inference endpoint for Llama-3-8B running on a single H100 GPU, both with and without a TEE enabled. We measured first-token latency and overall throughput under single-request and high-

concurrency (64 requests) workloads. For clarity, each reported value in Table 4 is averaged over 32 requests at concurrency 1 and 2048 requests at concurrency 64. Warm-up phase is excluded. Both the client and the endpoint run on the same machine to minimize network variance, and TLS is enabled, so the measured overhead includes encryption.

**Analysis.** Unlike software methods, TEEs are not vulnerable to computational attacks, as trust is anchored in hardware. The security guarantee is cryptographic, not statistical. Our evaluation in Table 4 shows that this strong guarantee comes at a modest and practical cost. The overhead for first-token latency was 9-16%, while the drop in

| Metric | Concurrent requests # | TEE | no TEE | Overhead |
|---|---|---|---|---|
| First token latency (ms) | 1 | 71 | 65 | 9.23% |
|  | 64 | 79 | 68 | 16.18% |
| Overall throughput (token/s) | 1 | 99 | 117 | 15.38% |
|  | 64 | 943 | 971 | 2.88% |

Table 4: Performance comparison for a vLLM endpoint with/without TEE (Meta-Llama-3-8B-Instruct, single H100, TLS enabled).

overall throughput under a high-concurrency load was only 2.88%. This demonstrates that TEEs offer a deployable, high-assurance solution to the model substitution problem, effectively closing the verification gap left by software-only techniques.

## 5 DISCUSSION

Our systematic evaluation reveals a clear hierarchy of verification capabilities:

**Text-only methods are insufficient.** Methods that only look at output distributions (classifiers, identity prompts, MMD, benchmarks) either need many queries, get confounded by benign variance, or can be evaded with some level of quantized model substitution. These approaches can give a rough signal but do not provide strong integrity guarantees. More generally, limited information access forces auditors to rely on large sample sizes to identify weak statistical patterns.

**Metadata-based methods are fragile.** Logprobs offer more sensitivity: in principle, per-token likelihoods can reveal discrepancies with far fewer samples than text-only outputs. However, in practice, backend nondeterminism destroys this advantage: variance across frameworks, batching, and hardware is often larger than the effect of quantization or fine-tuning. TopLoc improves robustness by producing compact activation-based proofs that are less sensitive to such variability, but it has no guarantees under significantly different inference settings, and still requires local recomputation of $M_{\text{spec}}$, making it infeasible for proprietary deployments. In practice, both logprobs and TopLoc collapse without access to the reference model.

**Hardware-backed attestation is strongest.** Trusted Execution Environments (TEEs) avoid these weaknesses: they guarantee integrity without exposing internals or requiring the auditor to hold the weights. Overheads are modest (Table 4), and attestation cryptographically binds the stack and model hash to the outputs. Adoption remains limited by operational complexity and weak provider incentives, but TEEs are currently the best compromise between verifiability, efficiency, and protection of proprietary models.

**Actionable recommendations:** (1) **For users:** Request attestation proofs when available; for critical applications, prefer providers offering TEE-backed services. (2) **For API providers:** Consider TEE deployment for premium tiers; increase transparency through model versioning and metadata disclosure. (3) **For policymakers:** Develop standards for API transparency and consider requiring attestation for regulated applications.

## 6 CONCLUSION

The model substitution problem represents a fundamental challenge in the current LLM ecosystem, where economic incentives misalign with user trust. Our work provides both a comprehensive analysis of why existing approaches fail and a concrete path forward through hardware attestation. As LLM APIs become increasingly critical infrastructure, ensuring computational integrity through TEEs may become as essential as HTTPS was for web security.

ETHICS STATEMENT

This study focuses on auditing techniques for detecting model substitution in LLM APIs. All experiments awere conducted using public datasets and benchmarks without involving human subjects or personal data, and API queries awere issued within the provider's terms of service and rate limits. The attacks we discuss against text-output–based verification are not presented to encourage misuse, but to motivate the research community to develop stronger auditing methods under the same level of API access. Our goal is to improve transparency and trust in LLM services by highlighting weaknesses in current auditing approaches.

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

## USE OF LARGE LANGUAGE MODELS (LLMS)

We acknowledge the use of LLMs as general-purpose assist tools during the preparation of this manuscript. Specifically, an LLM was utilized for paraphrasing certain sentences and for correcting grammatical errors and improving sentence structure throughout the text. This assistance was primarily aimed at enhancing the clarity and readability of our writing. The authors take full responsibility for the content of this paper, including any generated text, and confirm that all scientific ideas, experimental design, results, and conclusions are original contributions of the human authors. LLMs were not considered as authors or contributors to the research ideation.

## A  ADDITIONAL DETAILS

| Service Provider | Reference documentation |
| --- | --- |
| Anyscale | https://docs.anyscale.com/endpoints/text-generation/logprobs/ |
| Together.ai | https://docs.together.ai/docs/logprobs |
| Hugging Face | https://huggingface.co/docs/api-inference/tasks/chat-completion#request |
| AWS Bedrock | https://docs.aws.amazon.com/bedrock/latest/userguide/model-parameters.html |
| Nebius AI | https://docs.nebius.com/studio/inference |
| Vertex AI | https://cloud.google.com/vertex-ai/generative-ai/docs/multimodal/content-generation-parameters |
| Mistral | https://docs.mistral.ai/api/#operation/createChatCompletion |
| DeepSeek | https://api-docs.deepseek.com/api/create-completion |
| OpenAI | https://platform.openai.com/docs/api-reference/chat/create#chat-create-top_logprobs |
| Cohere | https://docs.cohere.com/v2/reference/chat#request.body.logprobs |
| Anthropic | https://docs.anthropic.com/en/api/messages |

Table 5: LLM API service providers documentations.

## B  QUANTIZATION SETTINGS

In all experiments we evaluate both INT8 (W8A16) and FP8 mixed-precision variants using off-the-shelf quantized models released by Neural Magic (e.g., `*-quantized.w8a16` and `*-FP8`). These models use the framework's default post-training calibration without modification.

## C  MODEL STEALING AND EMBEDDING VERIFICATION

**Background.** Researchers have explored extracting model information (e.g., weights, architecture details) from black-box APIs (Carlini et al., 2024; Finlayson et al., 2024). While successful extraction could reveal substitutions, these methods often require a vast number of queries, significant computational resources, and may still yield incomplete information, making them impractical for routine auditing by typical users. Finlayson et al. (2024) specifically demonstrated how logits analysis could expose model details, which we consider as a potential verification method for detecting model substitutions.

**Method.** The embedding size detection relies on the fact that logits generated by an LLM are restricted to a $d$-dimensional subspace of the full $v$-dimensional output space where $d$ is the hidden embedding size. So the auditor can analyze a set of logit outputs from different prompts and reconstruct this subspace and the hidden embedding size.

Given a set of $n$ output logit vectors $\{\ell_i\}_{i=1}^n$ (or log probabilities, since it preserves linear relationship up to a constant addition), we construct a logit matrix $L$ with each vectors as its columns. Applying SVD to $L$ yields $L = U\Sigma V^T$, where $U \in \mathbb{R}^{v \times v}$, $\Sigma \in \mathbb{R}^{v \times n}$ (with singular values $\lambda_1 \geq \lambda_2 \geq \ldots$), and $V \in \mathbb{R}^{n \times n}$. The magnitude of the singular values will drastically decreases after the first $d$ dimensions. Formally, the embedding size $d$ can be identified by detecting the singular value index at which the magnitude drops the most: $d = \arg\max_i(\log \lambda_i - \log \lambda_{i+1})$, Once identified, the $d$-dimensional subspace is constructed using the first $d$ left singular vectors from $U$: $U_d = [u_1, u_2, \ldots, u_d] \in \mathbb{R}^{v \times d}$, forming a unique signature of the model. Empirical evaluations confirm that this method accurately recovers the embedding dimensions. Furthermore, monitoring this subspace allows auditors to detect subtle changes, such as hidden system prompts, fine-tuning, or entire model substitutions from the service provider.

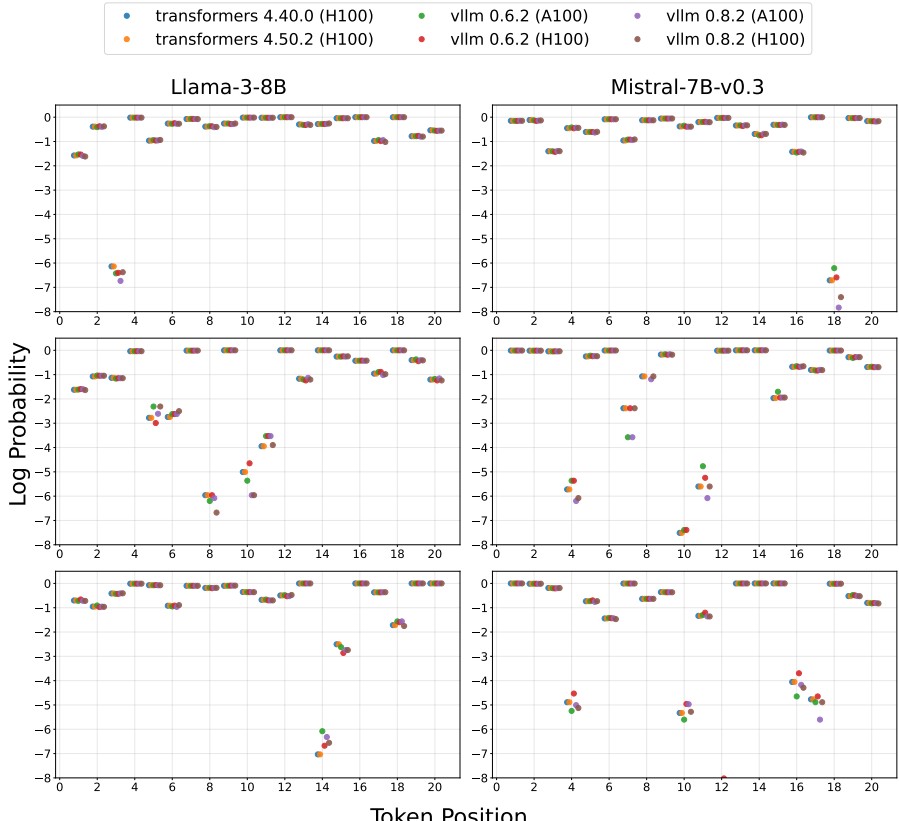

Figure 6: Log probability of generating first 20 shared tokens under greedy decoding for UltraChat Queries under different environment.

**Weakness.** This verification method is impractical because no well-known service provider gives full log probability access.

## D  BUILD AN LLM INFERENCE API ENDPOINT WITH TEE

In this section, we provide a solution for building an LLM inference API endpoint with TEE. Considering the main program is still running on the CPU, the GPU TEE technology needs to be paired with the VM-based CPU TEE (e.g., Intel TDX (Intel, 2021), AMD SEV-SNP (AMD, 2016)) to work properly. The latest CPU TEE can launch a confidential virtual machine inside and ensure the integrity and confidentiality of the VM's memory, preventing the host which is controlled by the API provider from manipulating the data inside the confidential VM. The GPU TEE (e.g., NVIDIA Confidential Computing (NVIDIA, 2023)) can build a secure communication channel between the CPU and GPU and ensure the integrity and confidentiality of the data on the GPU memory, thus we can add the GPU to the trusted computing base (TCB). By combining these two TEEs, we can ensure the integrity and confidentiality of programs and data processed inside them.

To ensure integrity and confidentiality, the program inside the TEE needs to generate a key pair for identifying itself and establishing secure communication channels with end users. This key can help end users verify that they are connecting to the program inside the TEE and ensure the API providers cannot interfere the communication traffic. Once the data is transferred to TEE, the data processing inside TEE will be encrypted by hardware to ensure integrity and confidentiality.

One last question is how we can provide proof of the integrity of the inference program and add this proof to the attestation report. First, the VM-based CPU TEE ensures memory protection and the measured direct boot (Murik & Franke, 2021) extends the measurement to kernel, initrd, and kernel command line. At this point, the measurement value in the attestation report can validate the

Figure 7: A screenshot of the API inference endpoint with TEE.

integrity of the kernel, initrd, and kernel cmdline. However, ensuring the integrity of the operating system and program on the disk requires additional steps. To achieve this, we include a program into the initrd that verifies and encrypts disk data. This action incorporates the disk into the trusted computing base (TCB).

In this setting, the LLM inference program needs to be open-sourced for verification, while the model can remain private if needed. The inference program inside the TEE can verify the hash of the model weights and ensure that every time a model with a specific name is used, it has the same hash. For open-sourced models, it is easy for anyone to calculate the desired hash. For proprietary models, the name serves as an alias for specific model weights, if we can confirm that it always has the same hash value, we can consider it to be integrous.

As we can see in Figure 7, the program is transparent and verifiable to the end users. End users can check the attestation report and verify the signature on it to ensure that the API provider is running the desired program.

