# OpenReview forum: "Are You Getting What You Pay For? Auditing Model Substitution in LLM APIs"
_ICLR.cc/2026/Conference — Submitted to ICLR 2026_

### Official Review · Reviewer_pbz9 · 2025-10-24

**Soundness:** 3
**Presentation:** 3
**Contribution:** 3
**Rating:** 6
**Confidence:** 3

**Summary:**

This paper addresses the critical issue of \emph{model substitution in commercial LLM APIs}---cases where service providers covertly replace an advertised model with a cheaper or quantized variant. The authors (1) formalize the model substitution problem and threat model, (2) evaluate existing software-only auditing methods such as text classifiers, MMD-based statistical tests, benchmark probing, and log-probability comparisons, and (3) propose \emph{Trusted Execution Environments (TEEs)} as a cryptographically secure and deployable solution. Through extensive experiments, they demonstrate that software-level approaches are unreliable in realistic, non-deterministic production settings, whereas TEEs provide provable model integrity guarantees with modest overhead (≈9–16\% latency). The paper concludes that TEEs are currently the most practical solution to ensure users “get what they pay for.”

**Strengths:**

- **Timely and relevant:** The paper targets a real trust gap in today’s LLM API ecosystem: users are billed for “Model A” but may actually get a quantized or smaller substitute.
- **Thorough evaluation:** The paper tests multiple auditing approaches under realistic adversarial settings:
  - text classifiers fail to distinguish full-precision vs. quantized variants (≈50% accuracy, i.e. random);
  - MMD tests lose power when the provider routes only part of traffic to the substitute;
  - benchmark-based auditing is confounded by hidden decoding parameters and caching;
  - greedy decoding and logprob comparison are broken by inference nondeterminism across stacks and batching.
- **Clear negative result:** It convincingly argues that software-only auditing is fundamentally brittle in production.
- **Actionable answer:** TEEs are proposed as a concrete, deployable alternative. The paper reports only ~9–16% first-token latency overhead and small throughput impact, which supports feasibility.
- **Responsible positioning:** The authors discuss implications for users, providers, and policymakers, not just measurements.

**Weaknesses:**

- **Novelty is mostly integrative:** The paper does not introduce a new detection algorithm or new attestation primitive. The contribution is primarily empirical and architectural.
- **Limited attack surface coverage:** The evaluation focuses on quantization, routing/mixing, and prompt caching. Other realistic substitutions (light fine-tuning on domain data, pruning, speculative decoding with a small draft model, etc.) are mentioned but not deeply quantified.
- **Security depth of TEEs:** The paper treats TEEs as “the answer,” but does not analyze:
  - side-channel leakage,
  - malicious firmware / compromised root of trust,
  - dishonest attestation endpoints in multi-tenant datacenters.
  These are important if we’re going to claim TEEs “solve” the problem.
- **Scalability and ops questions:** The TEE measurements are on a single GPU. It’s unclear how this extends to multi-GPU inference, pipeline parallelism, or high-throughput commercial routing.
- **Reproducibility / accessibility:** Verifying the TEE claims requires specialized hardware (confidential H100 / Blackwell stacks). That limits community validation.

**Questions:**

1. How does your auditing approach behave under *partial* substitution (e.g. 10–20% of requests silently routed to a cheaper model) when the attacker also slightly perturbs temperature or top-p to inject noise?
2. Can the proposed TEE-based workflow support multi-GPU inference or model sharding? If different GPU enclaves each serve a shard, how is attestation composed?
3. Could lightweight cryptographic proofs (e.g. ZKPs on selective layers, or TopLoc-style activation hashes) be combined with TEEs to reduce trust in the hardware vendor?
4. For open-source models that customers can run locally: do you see a practical “self-audit mode” that does not require TEEs?
5. Could the authors clarify the quantization settings used (e.g., INT8 vs FP8 schemes, calibration methods) for Table 2 and Figure 1?
   Were the same prompts and decoding parameters kept constant across quantized and full-precision variants?

6. In Figure 1, how sensitive are the MMD results to kernel choice and sample size?
   Did you observe any instability in power estimates under different prompt distributions?

7. For Table 4, how many runs were averaged to compute the reported latency and throughput overheads, and were warm-up tokens or network effects included in those measurements?

---

> ### Author Response · Authors · 2025-11-23
>
> Thank you for taking the time to read and carefully assess our work. We provide point-by-point clarifications below.
>
>
> ### **1. Novelty is primarily integrative rather than algorithmic**
> > “Novelty is mostly integrative: The paper does not introduce a new detection algorithm or new attestation primitive. The contribution is primarily empirical and architectural.”
>
> We agree that we do not introduce a new cryptographic primitive or statistical test. Our goal is instead to provide a security-and-systems study that
> - **stress-tests existing auditing proposals** under economically motivated substitution attacks, and
> - **demonstrates that TEE-backed inference is practically viable** for this audit task.
>
> By showing where software-only methods fail in adversarial, production-like settings and by quantifying the overhead of a hardware-rooted alternative, we shift the discussion from designing another detector toward **architectures capable of supporting contractual guarantees**. The novelty is therefore architectural rather than algorithmic.
>
>
> ### **2. Limited attack-surface coverage**
> > “The evaluation focuses on quantization, routing/mixing, and prompt caching. Other realistic substitutions (light fine-tuning, pruning, speculative decoding, etc.) are mentioned but not deeply quantified.”
>
> Quantization is a useful representative of model-substitution attacks because it introduces smaller and more subtle weight-level changes, making it strictly harder to detect and giving attackers strong economic incentives to use it. Fine-tuning is also a weight modification and would count as substitution, but it is generally easier to detect and providers have weaker incentives to rely on it. For inference-time substitutions such as pruning-based routing or speculative decoding, a full evaluation requires provider-side infrastructure; these are therefore natural extensions and part of our future work scope.
>
> ### **3. Security depth of TEEs**
> > “The paper treats TEEs as ‘the answer’ but does not analyze side-channel leakage, malicious firmware, compromised root of trust, or dishonest attestation endpoints in multi-tenant datacenters.”
>
> Risks such as side-channel leakage, malicious firmware, or a compromised root of trust are **standard assumptions in confidential computing** and lie outside the auditor’s control. Importantly, the auditor does **not** rely on a provider-side attestation endpoint: users verify the signed attestation report themselves. We now make these trust assumptions explicit in the revision.
>
> ### **4. Scalability and operational considerations**
> > “The TEE measurements are on a single GPU. It’s unclear how this extends to multi-GPU inference, sharding, or high-throughput routing.”
>
> Our measurements are limited to a single GPU due to compute constraints. However, NVIDIA’s latest confidential-compute stack supports multi-GPU TEEs, enabling sharded or pipelined inference **within one attested environment**. Because most overhead arises at enclave setup, we expect the incremental multi-GPU cost to be modest. Existing confidential-inference services (e.g., Phala) further demonstrate practical feasibility. We've also added this clarification to the paper.
>
>
> ### **5. Reproducibility and accessibility**
> > “Verifying the TEE claims requires specialized hardware (confidential H100 / Blackwell). This limits community validation.”
>
> We agree. Hardware-rooted security inherently requires specialized devices, so reproducibility is unavoidably more limited than for software-only auditing approaches. We now state this constraint directly.
>
>
> ### **6. Partial substitution with noise**
> > “How does your auditing approach behave under partial substitution (e.g. 10–20% of requests) when the attacker perturbs decoding parameters?”
>
> Under partial substitution (e.g., 10–20%), both MMD-based and benchmark-based audits **quickly lose power** and approach the significance level. We think that small perturbations to decoding parameters (temperature, top-p) will further obscure the signal and still align with the result that **software-only audits are brittle under mild adversarial conditions**.
>
>
> ### **7. Multi-GPU composition**
> > “Can the proposed TEE-based workflow support multi-GPU inference or model sharding? If different GPU enclaves serve shards, how is attestation composed?”
>
> NVIDIA’s upcoming confidential-compute support allows multi-GPU inference **inside a single attested environment**, so the attestation report covers the combined execution context. No manual composition of multiple enclaves is required.
>
> (1/2 To be continued)

---

> ### Author Response · Authors · 2025-11-23
>
> (2/2)
>
> ### **8. Combining TEEs with lightweight proofs**
> > “Could lightweight cryptographic proofs (e.g., ZKPs on selective layers, or TopLoc-style activation hashes) be combined with TEEs to reduce reliance on the hardware vendor?”
>
> Yes. Techniques such as partial ZK proofs or activation hashing can complement TEEs by reducing reliance on the hardware vendor.
>
>
> ### **9. ‘Self-audit mode’ for open-source models**
> > “For open-source models that customers can run locally: do you see a practical ‘self-audit mode’ that does not require TEEs?”
>
> For open-source models, users could in principle compare local logprobs or greedy outputs to those served by a provider. In practice, **inference nondeterminism and small decoding-parameter mismatches make this comparison unstable** over APIs, so we do not view this as a reliable audit mechanism.
>
>
> ### **10. Clarification of quantization settings**
> > “Could the authors clarify the quantization settings used (INT8 vs FP8 schemes, calibration methods)? Were the same prompts and decoding parameters kept constant across quantized and full-precision variants?”
>
> Prompts and decoding parameters are held constant across full-precision and quantized variants. Our experiments use off-the-shelf FP8 and INT8 (W8A16) models from Neural Magic, relying on their default post-training calibration without modification. We added a clarification on quantization settings also in the appendix.
>
> ### **11. MMD sensitivity to kernel choice and sample size**
> > “In Figure 1, how sensitive are the MMD results to kernel choice and sample size? Did you observe any instability across prompt distributions?”
>
> We did not run separate prompt-distribution experiments. Prior work (e.g., Gao et al.) reports minimal prompt-dependent instability, and our observations are consistent with this: **MMD power is dominated by kernel choice and sample size** rather than prompt selection.
>
> ---
> Thank you again for your thoughtful feedback. Your comments helped clarify the scope of our claims and strengthened the presentation of our results.

---

### Official Review · Reviewer_gV7d · 2025-10-27

**Soundness:** 3
**Presentation:** 2
**Contribution:** 2
**Rating:** 2
**Confidence:** 3

**Summary:**

This paper studies the problem of auditing model substitutions in LLM APIs. It formalizes the hypothesis test and evaluate a series of techniques, including text-only tests, log-probability-based tests, activation-based tests. They find these methods all fail under at least one type of adversarial substitution attack, and they claim that trusted execution environments provide the only robust guarantee for solving the problem.

**Strengths:**

1. Important problem and clear motivation: The problem they study is timely and important, given the widespread use of blackbox LLM APIs.
2. Comprehensive empirical experiments: the paper systematically tests several verification strategies with different types of substitution attacks. The negative results are informative for future work.

**Weaknesses:**

### Major weaknesses

1. Definition of the auditing goal is too strong: the null hypothesis $H_0$ requires equality for all  (end of Section 3.1), which seems unattainable in realistic settings; the paper itself later also attributes failures of some methods to "production-level inference nondeterminism." Relaxed formulation of the hypotheses would be more coherent. However, with relaxed formulation, some substitution methods may no longer make sense, as discussed in the next point.
2. Quantization attack framing: the paper shows failure of most methods with quantization attacks. However, it is unclear from the text whether existing providers claim the dtype of their models, which makes "quantization substitution" potentially indistinguishable from allowed, valid, and legal backend optimization. Moreover, because benchmark performance deltas under quantization are small (as shown in Table 3), escaping detection may be both easy and practically operationally benign. More clarification on the motivation for this substitution method is needed. For example, if replacing model A with B improves provider-side cost-effectiveness but does not hurt user experience in any empirically observed scenario, do the authors still aim to detect this substitution? Also, it would be beneficial to consider the "severity" of constitution by how much they impact model performance, and then consider evaluation metrics for detection methods that take this "severity" into account.
3. TEE section lack details and depth: the paper claims that TEE is "the only currently viable mechanism" and lists this as one of their main contribution. However, TEE is only discussed at the very end of the paper with a short subsection. What exactly is attested with TEE? Does this include model weights and inference hyperparameters? How is the trust in this attestation verified by end-user? When I read up to the end of section one, I thought TEE as an auditing method would be a significant part of the main text.
4. Novelty is limited: the paper feels more like a position or survey paper. Much of section 4 implements known detecting techniques. If I understand it correctly, the main new contribution (as claimed by the authors) is the argument for using the existing TEE as a auditing method and the overhead measurement. This is valuable engineering evaluation but seems methodologically thin for ICLR.

### Minor issues
1. Adversary model is too narrow: the randomized substitution attack mixes models uniformly randomly, but strategic provider would very likely condition their substitution decisions on certain features of the request. The paper does not evaluate adaptive substitution. which is arguably the more realistic case.
2. Benchmark-based detection seems brittle: it seems that the adversary can simply return fixed outputs for all known benchmark queries.

**Questions:**

## Questions
Many questions are already discussed in the weaknesses section. Some selected/additional questions are:
1. Would you consider relaxing the null hypothesis? If so, how? How does this relaxation change your conclusions?
2. How do you propose distinguishing "benign" quantization (operationally equivalent for users) from substitutions that meaningfully violate a service agreement?
3. Is the production-level deployment of TEE practical? What are some potential blockers? How can a user verify that every request was served by the TEE?

---

> ### Author Response · Authors · 2025-11-23
>
> Thank you for the detailed and thoughtful review. Below we address each of your concerns in turn.
>
> ### **1. Strength of the auditing hypothesis**
> > “Definition of the auditing goal is too strong… equality for all seems unattainable… relaxed hypothesis would be more coherent.”
>
> We agree that the full token output distribution equality is too strong. We have updated the hypothesis in the paper to focus on **model weight-level substitution** rather than equality of full output distributions. This aligns the hypothesis with our actual audit goal and does not change our main conclusions.
>
>
> ### **2. Quantization attack framing**
> > “Unclear whether dtype is advertised; quantization may be indistinguishable from valid backend optimization… benchmark deltas are small… why treat quantization as substitution?”
>
> We include quantization as a substitution case because the advertised service is tied to a **specific model checkpoint**; any undisclosed transformation of the weights—quantization or finetuning—modifies that checkpoint and is therefore within the scope of substitution an auditor may wish to detect.
>
> Although average benchmark differences can be small, quantization is known to affect performance in certain domains (e.g., code generation and multi-step reasoning). Our goal is not to claim that all quantized variants are harmful, but that providers should be honest about undisclosed alterations. Severity measures are a reasonable extension; our study focuses on detectability, and such metrics can be added to quantify the size of any detected deviation.
>
> ---
>
> ### **3. TEE section depth and clarity**
> > “TEE is claimed to be ‘the only viable mechanism’ but appears only at the end… What exactly is attested? How does user verify trust?”
>
> We have included the TEE section by summarizing the full workflow described in Appendix C.
>
> The attestation covers:
> - the CPU and GPU confidential-compute environment,
> - the LLM inference program,
> - the verification code that checks a hash of the model weights.
>
> In our setting, the inference program is open-sourced so users can verify its integrity, while the model weights may remain private: the provider publishes a reference weight hash and users check that the attested hash matches this value. This clarifies what is attested and how end users validate trust in the deployed model.
>
> ---
> ### **4. Novelty considerations**
> > “Novelty is limited: the paper feels more like a position or survey paper. Much of section 4 implements known detecting techniques. If I understand it correctly, the main new contribution (as claimed by the authors) is the argument for using the existing TEE as an auditing method and the overhead measurement. This is valuable engineering evaluation but seems methodologically thin for ICLR.”
>
> We agree that our work builds on existing statistical tests and hardware primitives, but we believe the contribution is more than a position or survey. Our experiments provide, to our knowledge, the first systematic evidence that SOTA software-only methods (classifiers, MMD) that succeed at model identification fail under subtle weight-level substitution and realistic decoding noise. We also provide the first evaluation of TEE-based LLM inference specifically for substitution auditing, showing that the overhead is small enough for practical deployment. Together, these results support a concrete architectural claim about what is and is not achievable in this setting.
>
>
> ### **5. Adversary model: randomized substitution**
> > “Randomized substitution is too narrow; a strategic provider would condition substitution on request features… adaptive substitution not evaluated.”
>
> We agree that adaptive substitution is realistic. We discussed benchmark-evasion attacks where the provider detects audit prompts and returns cached outputs; simulating more complex, traffic-dependent routing is not feasible without real production data. Randomized substitution therefore serves as a **lower-bound** attack that already shows the brittleness of benchmark-based and other software-only auditing methods.
>
>
> ### **6. Benchmark-based detection brittleness**
> > “Adversary can simply return fixed outputs for all known benchmark queries.”
>
> We agree. This is precisely why the paper argues that benchmark-based detection is brittle and provides an explicit example (benchmark evasion via caching) as part of our proposed attacks. However it is trivial to evaluate exact prompt level string-match so we did not include any results.
>
> ---
>
> ## **Questions**
>
> ### **7. Relaxing the null hypothesis**
> > “Would you consider relaxing the null hypothesis? How would this change your conclusions?”
>
> Yes. We have updated the hypothesis to focus on **model weight-level substitution**, which reflects the actual contractual claim. This does not change our main conclusions.
>
> (1/2 To be continued)

---

> ### Author Response · Authors · 2025-11-23
>
> (2/2)
>
> ### **8. Distinguishing ‘benign’ quantization**
> > “How do you propose distinguishing benign quantization from harmful substitution?”
>
> If a provider advertises a specific model checkpoint, any undisclosed modification of the weights constitutes substitution, even if average benchmark performance appears similar. Our goal is not to classify such changes as harmless or harmful, but to ensure that undisclosed alterations remain detectable.
>
>
> ### **9. Practicality of TEE deployment**
> > “Is TEE-based deployment practical? How can a user verify every request was served by the TEE?”
>
> Confidential-compute platforms such as **Phala Cloud** already use TEEs to serve LLM inference on H100-class GPUs. In these systems, the TEE provides an attestation report covering the hardware, OS, and source code. Users verify the attestation report and the signature or public key of the API endpoints, ensuring that requests are served by the attested TEEs.
>
> ---
>
> Thank you again for the thoughtful and detailed feedback! It helped us refine the substitution scope and able to clarify the details.

---

### Official Review · Reviewer_4MTF · 2025-10-28

**Soundness:** 3
**Presentation:** 3
**Contribution:** 3
**Rating:** 6
**Confidence:** 4

**Summary:**

This paper addresses the critical trust problem in commercial LLM APIs where users pay for specific models but cannot verify if providers are faithfully serving them. The authors formalize the model substitution detection problem, systematically evaluate existing software-based detection methods under adversarial conditions, and propose Trusted Execution Environments (TEEs) as a robust hardware-based solution. Through comprehensive empirical analysis across multiple detection techniques (text classifiers, identity prompting, statistical tests, log probability verification), they demonstrate that software-only approaches are fundamentally unreliable due to inference nondeterminism and subtle substitution strategies, while TEEs provide cryptographic guarantees with modest performance overhead (2.88-16.18%).

**Strengths:**

This paper makes several important contributions to addressing the critical trust problem in commercial LLM APIs. Most notably, it tackles a timely and economically significant issue where providers have strong incentives to substitute cheaper models while maintaining premium pricing, creating a fundamental trust gap in the current ecosystem. The authors provide a comprehensive and systematic evaluation of detection methods that goes well beyond previous work by considering realistic adversarial scenarios including quantization substitution, randomized model mixing, and benchmark evasion attacks. Their empirical analysis reveals crucial practical insights, particularly the surprising finding that production-level inference nondeterminism—arising from factors like batch variance and heterogeneous backends—can defeat seemingly robust detection methods such as log probability verification, even when these methods work well in controlled laboratory settings.

The paper's technical approach demonstrates strong rigor through its formal problem definition using a hypothesis testing framework (H₀: Pactual = Pspec vs H₁: Pactual ≠ Pspec) that properly grounds the empirical evaluation. The experimental methodology is sound, employing appropriate statistical tests like Maximum Mean Discrepancy with permutation testing, and the results are clearly presented through effective visualizations that illustrate key findings such as the relationship between substitution probability and detection power. Perhaps most importantly, the paper goes beyond merely identifying problems to propose and validate a practical solution through Trusted Execution Environments (TEEs). The authors demonstrate that TEEs can provide cryptographic guarantees of model integrity with surprisingly modest performance overhead—only 2.88% throughput reduction under high concurrency with 64 concurrent requests—making this a genuinely deployable solution for the industry. This combination of thorough problem analysis, systematic evaluation under adversarial conditions, practical insights about real-world deployment challenges, and an actionable solution with empirical validation makes this work a valuable contribution that directly addresses pressing concerns in the rapidly evolving LLM API ecosystem.

**Weaknesses:**

1. Limited TEE Evaluation: Only evaluates one model (Llama-3-8B) on one GPU (H100). Lacks testing on larger models (70B+) where memory constraints and multi-GPU setups could significantly impact feasibility and overhead. Since TEEs are proposed as the primary solution, this limited evaluation scope severely constrains confidence in the solution's general applicability.

2. Limited Real-World Validation: All experiments use controlled settings with known models and substitutions. The paper lacks evaluation with actual commercial APIs or evidence of real-world substitution detection. Without validation against production systems or case studies of actual substitution attempts, the practical applicability of both detection methods and the TEE solution remains uncertain.

3. Narrow Attack Scenarios: Focuses primarily on basic quantization and randomized substitution. Doesn't explore sophisticated adversarial strategies like adaptive substitution based on query patterns, gradual model degradation, or hybrid evasion techniques. This limits understanding of how robust the detection methods and TEE solution would be against determined adversaries with economic incentives to evade detection.

**Questions:**

1. How would the TEE solution scale to larger models (175B+ parameters) that require multi-GPU setups? What are the technical challenges and expected overheads?

2. Have you considered hybrid approaches combining multiple detection methods (e.g., periodic TEE attestation with continuous statistical monitoring)?

---

> ### Author Response · Authors · 2025-11-23
>
> We appreciate your careful and constructive review, and thank you for recognizing the contribution of our work. We address your main concerns and questions point by point below:
>
> ### **1. Limited TEE Evaluation**
> > “Only evaluates one model (Llama-3-8B) on one GPU (H100)… limited scope constrains confidence in general applicability.”
>
> Due to compute limitations, we were not able to run multi-GPU TEE for larger models. Nvidia’s documentation shows that with the latest drivers, **multi-GPU TEE is feasible.** Although we cannot test multi-GPU overhead directly, we expect only modest additional overhead. Existing confidential AI services (e.g., **Phala Cloud**) also demonstrate that TEE-based inference is feasible in practice.
>
>
> ### **2. Limited Real-World Validation**
> > “All experiments use controlled settings… lacks evaluation with commercial APIs or actual substitution attempts.”
>
> We agree this is a limitation. Evaluating real commercial APIs or confirmed substitution cases is complicated by the lack of publicly known incidents and the absence of ground-truth access to provider weights. Our study therefore focuses on controlled substitutions but identifies failure modes in current auditing methods that would apply equally in production.
>
> However, we added **Figure 5 (line 390)** and accompanying analysis showing that commercial APIs exhibit low inter-provider agreement and are not even consistent with themselves under greedy decoding. This shows that **output/distribution-based defenses are inherently unreliable in real-world settings**.  However, more detailed real-world validation remains valuable future work.
>
> ### **3. Narrow Attack Scenarios**
> > “Focuses on basic quantization and randomized substitution… does not explore adaptive or hybrid adversarial strategies.”
>
> We agree that our study focuses on relatively naive substitution strategies (quantization and randomized mixing). We also discuss a stronger benchmark-evasion scenario, where a malicious service provider routes benchmark queries to cached outputs. More adaptive attacks (e.g., query-pattern or traffic-based routing) are realistic but cannot be evaluated without access to real API traffic and production routing systems. Our chosen attacks therefore serve as **lower-bound examples** that already reveal key weaknesses in software-based auditing.
>
>
>
> ### **4. Hybrid approaches**
> > “Have you considered combining TEE attestation with continuous statistical monitoring?”
>
> Yes. Hybrid approaches are feasible: **TEEs provide the strongest guarantee** by cryptographically attesting the code and weights, and this signal is not affected by decoding noise or inference nondeterminism. **Statistical monitors can still provide complementary information** between attestations or if compliance is partial. We therefore view the two approaches as complementary rather than mutually exclusive.
>
> ---
>
> Thank you again for highlighting these points. We hope these clarifications and additional results address your concerns on practicability and scope.

---

### Official Review · Reviewer_eP5M · 2025-10-29

**Soundness:** 1
**Presentation:** 2
**Contribution:** 1
**Rating:** 2
**Confidence:** 4

**Summary:**

This paper studies the problem of auditing model substitution in LLM APIs. When users use an API provider to generate text from an LLM, they may want to check that they are being served the correct model that they are paying for. API providers may have an incentive to secretly use a different model, e.g., a smaller model to save on compute costs.

This paper runs experiments to claim that existing software-based auditing methods are unreliable in adversarial scenarios, such as quantization and randomized substitution. Therefore, this paper proposes trusted execution environments (TEEs) as a solution, where hardware enclaves provide cryptographic attestation reports of correct model and code execution.

**Strengths:**

The problem of auditing model substitution is well-motivated and important given the widespread usage of LLM APIs today. The introduction clearly and succinctly states the problem setting. The proposed use of TEEs for auditing model substitution appears to be a novel idea.

**Weaknesses:**

I don’t think that the paper has substantial enough contributions for what is typically expected in a publication. The majority of the paper consists of simple evaluations of existing methods on adversarial settings, such as subtle differences due to quantization. Note that several of these existing methods were originally designed for problems other than auditing model substitution. For example, [Sun et al. (2025)](https://arxiv.org/abs/2502.12150) studies the problem of predicting which model generated a particular text, e.g., ChatGPT vs Claude.

[Gao et al. (2024)](https://arxiv.org/abs/2410.20247) studies the same problem of auditing model substitution as the submitted paper, and proposes and evaluates principled, rigorous statistical tests (MMD) for doing so. Gao et al. (2024) showed strong results in controlled, simulated settings, and also ran audits on real-world LLM APIs. I am not convinced by the submitted paper’s claim that MMD is an unreliable and impractical audit method.

The submitted paper runs a small experiment claiming that MMD achieves low power for detecting quantization of small models and randomized substitution. However, the paper does not report how many completions are sampled for each prompt when running MMD, which is a crucial factor impacting the power, as found in Gao et al. (2024). I suspect that given more samples, MMD would achieve higher power.

Another recent arXiv preprint that proposes an audit for model substitution is [Zhu et al. (2025)](https://arxiv.org/abs/2506.06975), although I do not expect a detailed experimental comparison since it is a preprint and recent (June 2025).

The proposed TEE method does not seem practical for deployment in real-world APIs, given the complexity of modern LLM inference infrastructures. Larger models are split across many GPUs, and complex routing strategies are employed due to prompt caching, mixture of experts, etc. In addition, the proposed TEE method requires the open-sourcing of inference code, which would likely be unacceptable for both proprietary model providers and providers of open-source models. Such code would reveal information about model architecture, inference optimizations and tricks, etc.

Note that TEEs require API providers to make significant implementation changes, whereas audits such as MMD can be run by users on any black-box LLM API, without needing the API provider to agree to it. So TEEs are not a comparable replacement for black-box audits.

**Questions:**

### Main questions

1. Line 141: shouldn’t this be $p \\to 0$ for a low substitution rate? Then it should be $p$ upward on line 143\.
2. Line 246: doesn’t the prompt have to appear in the MMD equation? It only makes sense to compare completions that came from the same prompt.
   * It is confusing to use $x$ to denote completions here, as $x$ was used for prompts and $y$ for completions earlier in the paper.
   * The model distribution was only defined as the conditional distribution $P(y \\mid x)$ earlier in the paper, so it is not clear what exactly drawing $x \\sim P$ means here.
3. Figure 1 (left): why are there two bars for original and quantized? Comparing the original and quantized models is one test, so why isn’t there just one value for the power?
4. Line 253: how many completions are sampled for each prompt in the test? Gao et al. (2024) found that the power is sometimes low when few samples are used, but high when more samples are used.
5. Line 269: where does Gao et al. (2024) state that *“MMD-based auditing is only effective under strictly controlled local inference environments, limiting its practicality”*? Gao et al. (2024) run audits on real-world API providers, so I’m not sure where this is coming from.
6. Line 317: the size of the drop in accuracy for higher temperatures seems significantly higher than the drop from quantization. The drop from higher temperature is around 5–10 percentage points, whereas the difference from quantization is usually around 1–2 percentage points.
7. Can’t the effect of higher temperature be mitigated by taking the most common answer across queries, instead of averaging? The relative probabilities between answers should not change in higher temperatures.
8. Line 350 states that both $k \= 20$ and $k \= 100$ is used, but only one agreement rate is reported. Which value of $k$ is used?
9. In section 4.1.5, how different are the greedy decoding outputs when they do not match? Are they completely different, or just off by one or two tokens?
   * What is the agreement rate between providers, i.e., comparing Together vs. OpenRouter instead of Together vs. local baseline?
   * Are the greedy outputs deterministic if you query the provider with the same prompt multiple times, or do they vary?
10. Line 410: this definition of LSH is contradictory, how can “similar activations yield similar hashes” while also “small changes cause detectable differences”? LSH is designed so that similar items receive the same hash with probability, so the second part seems to be incorrect.

### Minor questions

1. Line 246: why does MMD have the squared superscript? It does not appear with the square elsewhere in the paper.
2. Line 252: what is $T$? I assume that $L$ is the completion length, but this is also not explicitly stated.
3. Line 252 states that Llama 70B is used, but Figure 1 (left) shows only Llama 8B and Mistral 7B.

---

> ### Author Response · Authors · 2025-11-23
>
> Thank you very much for the careful and detailed review. We respond to your main concerns and questions point by point below, and we have incorporated concrete fixes into the revised manuscript based on your comments.
>
> ---
>
> ### **1. Contribution and use of existing methods (classifiers)**
>
> > “I don’t think that the paper has substantial enough contributions… The majority of the paper consists of simple evaluations of existing methods on adversarial settings… these existing methods were originally designed for problems other than auditing model substitution. For example, Sun et al. (2025) study model identification (ChatGPT vs Claude).”
>
> We agree that prior classifier work (e.g., Sun et al.) [1] is designed for **model identification across families**, not for subtle **within-family weight-level substitutions**. Our contribution is to evaluate these methods in precisely this harder and practically motivated setting.
>
> - We treat **weight-level substitution** (e.g., quantization or randomized routing between variants of the *same* advertised model) as the audit target.
> - Under this setting, we find that model-ID classifiers are **largely insensitive to subtle substitutions** such as quantization, with accuracies near random.
> - To our knowledge, such behavior under a substitution threat model has not been systematically evaluated before, and our results show that these classifiers do **not** provide reliable evidence of small underlying weight changes.
>
> We have updated our section 3 —problem statement and threat model —to make our scope more clear.
>
>
>
> ### **2. MMD and what we mean by “practicality”**
>
> > “Gao et al. (2024)… propose rigorous statistical tests (MMD)… I am not convinced by the claim that MMD is unreliable or impractical.”
>
> Thank you for prompting us to clarify this. We now explicitly state that our **audit hypothesis focuses on weight-level substitution**, not general distributional shift.
>
> Under this framing:
>
> - Gao et al. [2] show that MMD can **detect distributional differences** and we observe this to a lesser extent in our quantization setting.
> - In real deployments, these differences can arise from **inference nondeterminism**, temperature/top-p variation, batching, and other unrelated factors.
>
> Our point is therefore specific: **MMD cannot reliably attribute observed differences to deliberate model substitution**, which is essential for contractual auditing. We revised the text to emphasize this *attribution limitation*, rather than portraying MMD as unreliable in general.
>
>
>
> ### **3. Sample size and statistical power for MMD**
>
> > “MMD achieves low power… the paper does not report the number of completions sampled… more samples may give higher power.”
>
> We agree, and this was underspecified earlier.
>
> - We now **explicitly report** the number of completions per prompt in all MMD experiments. (line 262)
> - Increasing sample size would indeed increase power, shown in Gao et al. (2024) [2].
>
> However, this introduces a **practical trade-off**: larger samples increase query cost and also make MMD more sensitive to small distributional shifts unrelated to weight-level substitution (e.g., non-determinism factors). We highlight this trade-off as a core limitation of MMD as a primary tool for substitution auditing.
>
>
>
> ### **4. Practicality of TEEs under multi-GPU and complex routing**
>
> > “The TEE method seems impractical… models are split across many GPUs… complex routing… requires open-sourcing inference code.”
>
> We appreciate these concerns and have clarified our assumptions.
>
> **Multi-GPU and routing feasibility.**
> NVIDIA’s latest confidential-compute stack (e.g., Blackwell drivers) now supports **multi-GPU TEEs**, enabling sharding, pipeline parallelism, and MoE-style routing inside a **single attested environment**. When routing logic does not involve privacy-sensitive data, it can also be run **outside** the enclave (e.g., a standard router dispatching to multiple TEE-backed inference workers). Alternatively, the routing program can be placed inside a **CPU TEE**. Existing confidential-AI providers (e.g., **Phala Cloud**) already deploy GPU TEEs for production LLM inference, demonstrating feasibility.
>
> **Inference code vs. weight secrecy.**
> Our proposal requires disclosure of the **inference code** so that the attested measurement can be independently verified and covert routing cannot occur. The **model weights do not need to be public**:
>
> - The attestation covers the inference program and the verification logic that checks a **hash of the model weights**.
> - Providers can keep weights proprietary while publishing the **reference hash** that users verify against the attested value.
> - If a provider prefers not to open-source the inference code, they may instead rely on a trusted third party to audit and certify it.
>
> (1/2 To be continued)

---

> ### Author Response · Authors · 2025-11-23
>
> (2/2)
>
> ### **5. Relation to Zhu et al. (2025)**
>
> > “Another recent preprint is Zhu et al. (2025)… relevant but recent.”
>
> Thank you for pointing this out. We now Cite Zhu et al. (2025) [3] in related work in our updated paper.
>
> ### **6. TEEs vs. black-box audits (MMD) as complementary tools**
>
> > “TEEs require major provider changes, while MMD can be run by users on any black-box API… TEEs are not a replacement.”
>
> We fully agree. TEEs and black-box audits address different needs and are **complementary**.
>
> - TEEs provide **provider-side attestation** for strong contractual guarantees about the exact model checkpoint.
> - Black-box methods like MMD are **lightweight and user-initiated**, requiring no provider cooperation.
>
> Our intended claim is that **provider-side mechanism (TEE or similar) is necessary when strong contractual guarantees are required**. Practically, we consider a hybrid auditing model to be optimal: TEEs for cryptographic integrity of weights/code, and black-box tests as complementary signals.
>
> ---
>
> ### **Questions**
>
> Thank you for the detailed line-level feedback. In the revised paper we have:
>
> 1. Corrected the substitution-rate notation on lines 147–149: $p$ as the substitution probability and its limiting cases.
> 2. Fixed the MMD formulation (line 249) by conditioning on prompts and explicitly defining the conditional distributions $P(z \mid x)$, $Q(z \mid x)$, and the sampled completions $z$ and $z'$.
> 3. Removed the unintended squared superscript on MMD.
> 4. Clarified the experimental setup (line 262) by specifying the number of completions per prompt $N$ and the prompt set used.
> 5. Added Llama-3 and Llama-3.1-70B to Figure 1 and removed the non-substitute model from the reported power results.
> 6. Revised the claim about Gao et al. (line 278) to note that MMD can flag commercial providers with differing output distributions, but its practical power is limited by inference nondeterminism.
> 7. Clarified the agreement-rate definition (line 364) by explicitly stating the value of $k$ used.
> 8. Added a brief description of how greedy outputs differ in real API inference provider, reported inter-provider vs. intra-provider agreement rates, and verified that repeated greedy queries are notdeterministic (Figure 4) even when $k = 20$.
> 9. Corrected the LSH description (line 450) to align with its definitions.
>
> ---
>
> Thank you again for the very detailed and constructive review — your feedback directly improved the completeness and clarity of the revised manuscript.
>
> ---
>
> [1] Sun et al. *Idiosyncrasies in Large Language Models.* arXiv preprint arXiv:2502.12150, 2025.
>
> [2] Gao, Liang, Guestrin. *Model Equality Testing: Which Model Is This API Serving?* arXiv preprint arXiv:2410.20247, 2024.
>
> [3] Zhu et al. *Auditing Black-Box LLM APIs with a Rank-Based Uniformity Test.* arXiv preprint arXiv:2506.06975, 2025.

---

### Author Response · Authors · 2025-11-23
**General Response: Clarifying Novelty and Contributions**

We thank the reviewers for their detailed feedback. A recurring concern (Reviewers eP5M, gV7d, pbz9) is that the paper utilizes existing statistical methods (MMD, classifiers) or hardware (TEEs) rather than proposing a new algorithmic primitive, leading to questions about novelty.

We respectfully argue that our contribution is not a "survey" or "integration," but rather a **security analysis that falsifies the prevailing assumption that software-only auditing is sufficient.** Our novelty lies in three specific areas:

**1. Moving from Model Identification to Adversarial Auditing:**
While previous works like Sun et al. (2025) and Gao et al. (2024) propose methods for identifying models or detecting distributional shifts, they largely operate under non-adversarial or distinct-family settings (e.g., distinguishing GPT-4 from Claude).
Our work is novel because we evaluate these methods against a **specific, economically motivated adversary** performing *weight-level substitution* (e.g., quantization or randomized routing within the same family).
*   **New Finding:** We empirically demonstrate that methods which work for model identification (classifiers) collapse to near-random accuracy (approx. 50%) against quantization attacks (Table 1), a result not previously established.
*   **New Finding:** We show that while MMD is theoretically sound, it is operationally brittle in production. We identify that inference nondeterminism and hidden decoding parameters (Figure 3) act as confounding variables that generate false positives, rendering MMD unreliable for verifying *contractual* model integrity.

**2. Establishing the "Impossibility" of Robust Black-Box Auditing:**
A core contribution of this paper is the systematic evidence that **software-only signals are insufficient** for high-assurance auditing. This is not a negative result, but a critical security finding. We show that the information theoretic gap in black-box APIs (limited samples + nondeterminism + adversarial routing) makes statistical guarantees impossible to achieve without an unacceptable false-positive rate. This shifts the research conversation from "which statistical test is best?" to "how do we anchor trust?"

**3. First Rigorous Evaluation of TEEs for LLM Auditing:**
While TEEs exist as a primitive, their application is limited. We provide the **concrete system evaluation** demonstrating that TEEs are a viable solution for this specific problem.
*   **New Finding:** We debunk the assumption that TEEs are too slow for production LLMs, proving that the throughput overhead is negligible (<3%) under high concurrency (Table 4).
*   This transforms TEEs from a theoretical cryptographic possibility into an empirically validated, actionable solution for the model substitution problem.

---

> ### Comment · Reviewer_eP5M · 2025-11-26
>
> Thank you to the authors for their responses and revisions. Some comments and concerns:
>
> 1. > We respectfully argue that our contribution is not a "survey" or "integration," but rather a security analysis that falsifies the prevailing assumption that software-only auditing is sufficient.
>
>    The paper only evaluates MMD with 10 completions per prompt, 25 prompts, so 250 samples total. This is the smallest number of samples used in the experiments in [Gao et al. (2024)](https://arxiv.org/abs/2410.20247), and they find that increasing the number of samples per completion can significantly increase the test power. To falsify that software-only auditing is sufficient, a fairer evaluation of these methods is required, rather than only showing hyperparameter choices that are known to have weaker performance.
>
>    In fact, [Gao et al. (2024)](https://arxiv.org/abs/2410.20247) run essentially the same experiment in their Figure 6 as this paper does in Figure 1 left (detecting int8 quantization on Llama 3/3.1 8B/70B and Mistral 7B). Except they vary the number of completions per prompt from 10 to 100, and find that more samples leads to power close to 1\.
>
> 2. > While previous works like Sun et al. (2025) and Gao et al. (2024) propose methods for identifying models or detecting distributional shifts, they largely operate under non-adversarial or distinct-family settings (e.g., distinguishing GPT-4 from Claude). Our work is novel because we evaluate these methods against a specific, economically motivated adversary performing weight-level substitution (e.g., quantization or randomized routing within the same family).
>
>    [Gao et al. (2024)](https://arxiv.org/abs/2410.20247) run extensive experiments on detecting quantization (see Section 4.1, Appendix C, Figures 5, 6, 7).
>
> 3. > New Finding: We show that while MMD is theoretically sound, it is operationally brittle in production. We identify that inference nondeterminism and hidden decoding parameters (Figure 3\) act as confounding variables that generate false positives, rendering MMD unreliable for verifying contractual model integrity.
>
>    I don’t follow how inference nondeterminism would undermine MMD when it is typically run with temperature 1\. Also, [Gao et al. (2024)](https://arxiv.org/abs/2410.20247) show that MMD can easily detect differences in decoding temperature (Appendix C.3, Table 12).
>
> 4. I agree with Reviewer gV7d that the TEE section does not seem to have much substance research-wise. To quote Reviewer qV7d,
>    > - TEE section lack details and depth: the paper claims that TEE is "the only currently viable mechanism" and lists this as one of their main contribution. However, TEE is only discussed at the very end of the paper with a short subsection. What exactly is attested with TEE? Does this include model weights and inference hyperparameters? How is the trust in this attestation verified by end-user? When I read up to the end of section one, I thought TEE as an auditing method would be a significant part of the main text.
>    >
>    >  - Novelty is limited: the paper feels more like a position or survey paper. Much of section 4 implements known detecting techniques. If I understand it correctly, the main new contribution (as claimed by the authors) is the argument for using the existing TEE as a auditing method and the overhead measurement. This is valuable engineering evaluation but seems methodologically thin for ICLR.

---

> ### Author Response · Authors · 2025-11-27
>
> Thank you for the thoughtful follow-up. We agree with your points regarding sample size and clarify our intended claims below.
>
> ### **Sample size and intention of our MMD evaluation**
> We agree that our MMD evaluation uses a limited sample size, and Gao et al. (2024) clearly show that increasing completions per prompt substantially increases power for detecting quantization. Our intention was not to argue that statistical testing cannot detect **distribution shift** under large samples. Instead, we aim to highlight that achieving these sample regimes is **economically impractical** for individual auditors, and becomes even more difficult once a provider uses mixture routing or partial substitution where the shift signal is intentionally diluted.
>
> ### **Distribution shift vs. attribution in real API settings**
> Gao et al.’s test is very effective at detecting distribution differences, including small temperature changes. However, our new results (Figure 4) show that **the same provider**, using **the same prompt** and **greedy decoding**, does not consistently agree on the first 20 tokens. This indicates that **inference-time nondeterminism alone introduces distribution shift** even when the underlying weights are unchanged. Under such conditions, any text-based kernel used in MMD (or other kernels proposed by Gao et al.) will correctly detect a shift, but the result becomes ambiguous: the observed difference may stem from nondeterminism rather than from weight substitution. Our point is therefore about **attribution**, not detectability.
>
> ### **Why software-only auditing remains brittle**
> Because real API deployments inherently exhibit nondeterministic decoding behavior, software-based auditing methods that rely on output distributions cannot reliably separate nondeterminism-induced drift from true weight-level substitution. This challenge persists even when sample sizes increase, since the mechanism generating distribution shift is not under the auditor’s control.
>
> ### **Motivation for TEE-based guarantees**
> This motivates our argument for TEEs when strong contractual guarantees are required. TEEs attest directly to the inference program and the weight hash, avoiding reliance on output distributions. Our evaluation shows that TEE-based verification can be deployed with small overhead, making it a practical provider-side mechanism for high-assurance auditing.

---

### Meta-Review · Area_Chair_F8Xs · 2026-01-07

**Summary:**

This paper addresses the problem of auditing model substitution in LLM APIs, where providers may covertly serve cheaper or quantized models instead of the advertised ones. The authors evaluate a range of existing software-based auditing methods under adversarial conditions and argue that Trusted Execution Environments (TEEs) provide the only robust solution, offering cryptographic guarantees of model integrity with modest performance overhead.

**Reviewer Concerns:**

- Scope of evaluation: The experiments are restricted to relatively small models (e.g., Llama-3-8B) and single-GPU setups. This leaves open questions about scalability to larger models (70B–175B+) and multi-GPU inference pipelines, which are common in practice.
- Practicality of TEEs: Reviewers expressed skepticism about the deployability of TEEs in real-world API infrastructures. Challenges include multi-GPU sharding, complex routing strategies, and the requirement to open-source inference code for attestation. These factors may limit adoption by providers.
- Comparison to prior work: Reviewer 1 highlighted that Gao et al. (2024) already studied model substitution auditing with rigorous statistical methods (MMD) and achieved strong results, including audits on real-world APIs. The submitted paper’s dismissal of MMD as unreliable was not fully convincing, especially given the lack of detail on sample sizes and kernel choices.
- Narrow adversary model: The paper focuses on quantization and randomized substitution but does not explore more adaptive or sophisticated strategies (e.g., conditional substitution, gradual degradation, hybrid evasion). This limits the generality of the conclusions.

**Reviewer Scores:**

Reviewer gV7d may have a chance to raise the score.

---

### Decision · Program_Chairs · 2026-01-26

Reject